# The Tribo-Fatigue Damage Transition and Mapping for Wheel Material under Rolling-Sliding Contact Condition

**DOI:** 10.3390/ma12244138

**Published:** 2019-12-10

**Authors:** Chenggang He, Jihua Liu, Wenjian Wang, Qiyue Liu

**Affiliations:** 1School of Railway Tracks and Transportation, Wuyi University, Jiangmen 529020, Chinaljh214913@163.com (J.L.); 2State Key Laboratory of Traction Power, Tribology Research Institute, Southwest Jiaotong University, Chengdu 610031, China

**Keywords:** wear, wheel material, fatigue cracks, tribo-fatigue damage

## Abstract

The purpose of this work is to construct a tribo-fatigue damage map of high-speed railway wheel material under different tangential forces and contact pressure conditions through JD-1 testing equipment. The results indicate that the wear rate of the wheel material varies with tangential force and contact pressure. The wear mapping of the wheel material is constructed and divided into three regions: slight wear, severe wear, and destructive wear, based on the wear rate under each test condition. With an increase in tangential force and contact pressure, the maximum crack length and average crack length of the wheel material increases. According to the surface damage morphologies and corresponding statistical results of average crack length of wheel material under each experiment condition, a tribo-fatigue damage map is constructed and divided into three regions: slight fatigue damage region, fatigue damage region, and severe fatigue damage region. Fatigue cracks initiate on the wheel specimen surface. Some cracks may propagate into material and fracture under cyclic rolling contact; some cracks may grow into inner material with a certain depth, and then turn toward the surface to form material flaking; some cracks may always propagate parallel to the wheel roller surface.

## 1. Introduction

High-speed and heavy-haul railways are the main development areas for improving the global railway transport capacity in recent years. However, in modern high-speed and heavy-haul railway networks, wear and fatigue damage for wheel materials are important factors that reduce the service life of the wheel and rail system, and increase the risk of train operation [1,2,3]. In addition, trains often run on straight lines and curved lines with different radii, and repeatedly accelerate or decelerate during operation. The service environment of the trains is very complicated, and various damage forms appear on the wheel treads, for example: polygons wear, flange wear, tread wear, rolling contact fatigue (RCF), etc. [4,5,6]. RCF damage is also referred to as surface fatigue, pitting, or flaking in the literature. The initiation site of RCF damage may be situated below, or on, the contact surface. The former case is designated as sub-surface RCF, with initiation sites at sub-surface defects, resulting in fairly irregularly shaped craters or spalls in the contact surface [7]. The latter case is called surface initiated RCF, and the basic features of surface induced spalling damage are arrow-shape, opening in the rolling direction, and crack initiation at the apex [8]. Spalling is one of the typical RCF failures of railway rails and wheels. If the wheel slides on the rail, then the white layers may be caused by frictional heat on a part of the wheel tread and/or rail tread. After that, the cracks initiate, and spalling occurs in the white layers due to cyclic rolling contact [9]. The impact loads generated by the sliding flat on a part of the wheel/rail tread and the white layers generated on a part of the wheel/rail tread are the two main causes of spalling damage [10,11].

Wheel treads with rail head contact and wheel flanges with rail gauge contact, are two typical wheel–rail contact types, resulting in different tangential forces on the wheel and rail contact interface [12]. The tangential force has an important effect on the wear and RCF damage of wheel and rail materials [13,14,15,16]. Three wear regimes have been proposed in previous works, according to a series of experimental studies under different slip ratios [17,18]: mild wear, severe wear, and catastrophic wear. Some researchers use numerical methods to predict the wheel and rail wear profile. For example, a comprehensive friction work model was proposed by Li et al. [19] to predict severe wear for wheel and rail. Braghin et al. [20] established a wheel wear prediction model according to the Tγ/A method. In a subsequent study, the Tγ/A method was used by Wang et al. [21] to develop wear information and identify wear mechanisms and damage transition of wheel/rail materials with a third body material present in the contact interface.

Tribo-fatigue is the science of wear, fatigue damage, and failure of load friction pair systems in machines and equipment [22]. There are many factors affecting tribo-fatigue, and the contact interface is continuously changing due to friction and wear, geometrical characteristics, and material properties. Recent advances in tribology involve the use of the fundamental concepts of thermodynamic entropy. In addition to general experimental research, Basaran and Wang et al. [23,24,25] proposed a thermodynamic approach, based on the cumulative entropy generation, to predict metal fatigue. They showed that fatigue degradation and entropy generation are intimately related, and the cumulative thermodynamic entropy of a material up to the fracture is a unique material parameter. Sosnovskiy et al. [26,27] proposed a generalized theory of evolution, based on the concept of tribo-fatigue entropy. That is, tribo-fatigue entropy is determined by the processes of damageability conditioned by thermodynamic and mechanical effects, which causes the change of states in any system.

As an important part of the train, the wheel not only needs to bear the weight of the train, but also needs to transmit the action forces of the wheel/rail contact interface. However, the wheel/rail contact, due to its complex system and strong nonlinear coupling, makes the wheel and rail run under strong friction and multi-environment coupling. Therefore, the wheel and rail are typical tribo-fatigue components. The wheel tread is subjected to strong friction coupling, which causes serious tribo-fatigue damage; thus, greatly reducing the reliability and safety of train operation.

In order to reduce the service cycle cost and ensure the safety of the train, it is of importance to explore the tribo-fatigue damage characteristics of the wheel material under the rolling-sliding condition. In this study, in order to construct a tribo-fatigue damage mapping and transitions of the wheel material in the rolling-sliding contact process, a series of wheel/rail rolling-sliding contact experiments were performed under different tangential forces and contact pressure conditions through JD-1 testing equipment. The wear and tribo-fatigue damage mechanisms and transitions were discussed in detail.

## 2. Materials and Methods 

The wheel/rail rolling-sliding contact experiments were completed by using the JD-1 wear apparatus (Chengdu Rongteng Technology Development Co., Ltd., Chengdu, China) at average room temperature (18–25 °C) and relative humidity (50–70%) conditions. In order to ensure that the process of wheel/rail contact in the laboratory was identical with that on site, the Hertz simulation criterion was used in the test. That is, the maximum contact stress of wheel/rail and the ratio of the long axis to short axis of the elliptical contact spot in the laboratory were set to be identical with those on site [13]. The pressure of the contact spot is distributed in a semi-ellipsoidal shape. The geometric sizes of the simulated wheel and rail are determined by means of the Hertz simulation rule, shown in Equation (1) and Equation (2).
(1)$${\left(q_{0}\right)}_{lab}={\left(q_{0}\right)}_{field}$$
(2)$${\left(a/b\right)}_{lab}={\left(a/b\right)}_{field}$$
where, (*q*_0_)_lab_ and (*q*_0_)_field_ are the maximum contact stresses in the laboratory and in the field, respectively; (*a*/*b*)_lab_ and (*a*/*b*)_field_ are the ratios of the semi-major axis to semi-minor axis of the contact ellipses between the wheel and rail in the laboratory and field, respectively. The detailed usage and capabilities have been described previously [13], and a schematic of the apparatus is shown in Figure 1.

The rail roller (lower disc) acts as the driving disc and the wheel roller (upper disc) acts as the brake disc, when simulating a decelerating wheelset. The wheel roller utilized in the experiment was cut from the new wheel tread material. The rail roller used in this experiment was made of real rail material. The chemical compositions and the main mechanical properties of the railway wheel and rail rollers were supplied directly by the manufacturer (Pangang Group Co., Ltd., Panzhihua, China) and provided in Table 1 [16]. Figure 2 shows a typical microstructure of the pearlitic railway wheel material. It is obvious from Table 1 that the carbon content of the wheel material was lower than at the eutectoid point, meaning that the sample exhibits a ferrite–pearlite microstructure (lamellae containing both ferrite and cementite).

In this study, based on the contact geometry of the rail and wheel specimens, normal forces of 750, 1300, 2060, and 3070 N were used to simulate the maximum contact pressure of approximately 1000, 1200, 1400, and 1600 MPa between the rail and wheel interface, according to the Hertzian contact theory. Under each contact pressure condition, the tangential forces between the rail and wheel specimens were 40, 110, 220, and 350 N, respectively. The rotational speed of the rail specimen was 95 r/min, and the number of cycles that the wheel specimen experienced was 10^6^. Meanwhile the corresponding slip ratios at the rolling-sliding contact are about 0.15%, 0.8%, 1.4%, and 2.1%, respectively. The experiment conditions used are shown in Table 2. Two tests for each pressure–tangential force configuration were completed.

The wheel roller was weighed via an electronic balance (measurement accuracy: 0.001 g, Shanghai Hengping Scientific Instrument Co., Ltd., Shanghai, China) before and after each experiment to calculate the mass loss (g). The wear rate of the wheel roller was measured by mass loss (g) per rolling distance (m). Sections were cut in the direction longitudinal to the rolling direction and mounted in resin, ground to 2500 grit, polished to 0.25 μm diamond, and etched with 3% nital. The worn surfaces and sub-surface damage were observed via optical microscope (OM, OLYMPUS, Tokyo, Japan) and scanning electron microscope (SEM, OLYMPUS, Tokyo, Japan).

## 3. Results

### 3.1. Wear Rate and Wear Regime Map

Figure 3 shows the effect of tangential force and contact pressure on the wear rate of the wheel specimen. It was found in Figure 3a, that as the tangential force increases, the wear rate of wheel roller increases significantly. When the contact pressure increased, the wear rate of the wheel roller increased (Figure 3b). Furthermore, the change in tangential force is more sensitive to the effect of the wear rate of the wheel under relatively high contact pressure conditions. That is, as the tangential force increases, the tendency of the increase in the wear rate of wheel roller becomes stronger under relatively high contact pressure conditions.

A wear mechanism map is an important and more complete method to illustrate wear behavior of a wear system, such as wheel and rail friction pairs. A wear regime map is identified via the wear map and is defined by tangential force and contact pressure [18,28]. Figure 4 shows the wear mechanism map of wheel material. The independent variables are tangential force and contact pressure, and the dependent variable (the numerical point on the figure) is the wear rate of the wheel roller (× 10^−6^ g/m). The wear rate of the wheel roller is a minimum of 2.457 × 10^−6^ g/m, under the condition of less tangential force and lower contact pressure; while the wear rate of the wheel specimen is up to 29.641 × 10^−6^ g/m, under the condition of larger tangential force and larger contact pressure. Referring to the method proposed by Lewis et al. [18,28,29], the wear mechanism map of wheel material can be divided into three regimes of slight wear, severe wear, and destructive wear, by analyzing the wear rate under each tangential force and contact pressure testing condition.

Furthermore, the wear rate of the wheel roller corresponding to the transition boundary between the slight wear and severe wear regimes was about 8.0 × 10^−6^ g/m, and the wear rate of the wheel roller corresponding to the transition boundary between the severe wear and destructive wear regimes was about 20.0 × 10^−6^ g/m. In the slight wear regime, the wear rate of wheel material increases slightly with the increase of tangential force and contact pressure. But as the tangential force and contact pressure increases, the wear rate of wheel material has a large change in the severe wear regime, and the wear rate increases sharply in the destructive wear regime.

### 3.2. Surface Damage Behavior and Tribo-Fatigue Damage Transition Map

In order to further study the effect of tangential force and contact pressure on the damage mechanism of wheel material, the related microscopic analysis is particularly important. Representative SEM micrographs of worn surfaces of wheel specimens, under different tangential forces and contact pressure conditions, are shown in Figure 5. It was seen that the surface damage mechanism and morphology of the wheel material were significantly affected by the tangential forces and contact pressures. When both the tangential force and contact pressure were relatively small, the damage mechanism and morphology on the wheel specimen surface was dominated by a combination of slight fatigue cracks and spalling damage. When the tangential force and contact pressure were increased to a large extent, the damage mechanism and morphology on the wheel specimen surface was dominated by a combination of severe fatigue cracks and peeling damage. In addition, the influence of tangential force on the surface damage mechanism of the wheel material is more obvious. Under the condition of less tangential force, the damage behavior on the surface of wheel specimen is mild (Figure 5a–c); while under the condition of large tangential force, the damage behavior on the surface is serious (Figure 5d–h).

Three different positions of the wheel roller under each testing condition were randomly selected as the samples for analyzing the fatigue damage, and then all the surface fatigue cracks on the cross section of each analytical sample were observed and photographed via OM. Then the crack propagation length and the crack propagation angle of every fatigue crack for each analytical sample was measured, and the average crack propagation length, the average crack propagation angle, and the corresponding standard deviation of the analytical sample for every test was calculated. The main purpose of this was to statistically analyze the rolling contact fatigue damage behavior of wheel material under different tangential forces and contact pressure conditions. Figure 6 only shows the OM micrographs of two representative surface fatigue cracks for the wheel specimen and a schematic diagram of the measurement of the crack length and crack angle of the wheel sample. A single layer crack (Figure 6a) and multi-layer cracks (Figure 6b) were observed on the longitudinal section along the rolling direction of wheel roller. Throughout the observation process, many single-layer cracks and multi-layer cracks were found on the wheel specimen, distributed along the rolling direction of the wheel specimen and propagated in different angles.

The statistical results of the crack length and crack angle of wheel rollers are shown in Figure 7. When the tangential force is constant, the maximum crack length and average crack length of the wheel material increases as the contact pressure increases (Figure 7a), while the maximum crack angle of the wheel specimen has no obvious change, and the average crack angle increases slightly, ranging between 8–12°. In addition, when the contact pressure is constant, the maximum crack length and average crack length of the wheel material generally increases gradually with the increase in tangential force (Figure 7a), but the maximum crack angle and average crack angle of the wheel sample also do not change significantly (Figure 7b).

The tribo-fatigue damage transition map can visually describe the changes of different damage forms for the wheel material under different tangential forces and contact pressure conditions. For the purpose of constructing the tribo-fatigue damage mechanism map of the wheel material, both surface damage morphology and average crack length are taken into consideration in this study.

According to the comprehensive analyses of the surface damage morphologies (Figure 5) under each tangential force and contact pressure condition and the corresponding statistical result of the average crack length (Figure 7), the tribo-fatigue damage mechanism map of the wheel material is constructed, as shown in Figure 8. The tribo-fatigue damage transition map of wheel material can be divided into three regions: slight fatigue damage, fatigue damage, and severe fatigue damage (Figure 8). The average crack length of the wheel roller corresponding to the transition boundary between the slight fatigue damage and fatigue damage regimes is about 60 μm, and the average crack length of the wheel roller corresponding to the transition boundary between the fatigue damage and severe fatigue damage regimes is about 100 μm. In general, the surface damage of the wheel specimen is mainly surface fatigue cracks. In the slight fatigue damage region, the surface damage of the wheel specimen is mild, the damage mechanism is slight fatigue damage and spalling, and the surface fatigue cracks are relatively small (Figure 5a,b). At the same time, the influence of contact pressure on the surface damage for wheel specimens is not obvious in slight fatigue damage region. In the fatigue damage region, the surface damage of the wheel specimen gradually intensifies; the damage mechanism is dominated by surface fatigue cracks and peeling, and the contact pressure has a certain influence on the surface damage morphology of wheel material (Figure 5c–f). In the severe fatigue damage region, the damage mechanism of the wheel specimen is mainly severe surface fatigue cracks and peeling (Figure 5g,h).

## 4. Discussion

It was found that the ferrite phase in wheel material was shaped into a line due to the plastic deformation from the representative photomicrograph of the cracks (Figure 9). The fatigue cracks initiate from the surface and subsurface of the wheel sample and propagate in the direction of plastic deformation. Single principle cracks, multi-layer cracks, branch cracks, and sub-surface cracks can be clearly observed from the section of the wheel roller under each tangential force and contact pressure condition (Figure 9). It appears that the single layer cracks occur relatively more often, and the multi-layer cracks occur relatively less often. In the multi-layer cracks, the inter-laminar material between the inner and outer cracks of the multi-layer cracks tends to rupture (Figure 9c). This is because the multi-layer cracks are easily crushed and broken by the cyclic contact stress. After the branch crack initiates from the principle crack, it propagates along the ferrite line in the plastic deformation area with a small angle (Figure 9d).

The cracks may change propagating direction after encountering resistance during the propagating process, shown in Figure 10 (OM and SEM photographs). The change of tangential force and contact pressure caused by the change of the contact state between wheel and rail rollers, and the complex fatigue cracks growth, led to the turning of the crack propagation direction during the propagating process. Therefore, there is a significant increase in the difficulty of predicting the crack propagation direction of wheel material. Surface cracks and/or sub-surface cracks with different propagating directions are likely to be connected to each other to form wear debris during the rolling-sliding contact process.

The microstructure of the wheel material is a ferrite–pearlite structure (Figure 2), and the ferrite phase is relatively soft. In the plastic deformation region, the ferrite structure is extruded into ferrite lines, and the pearlite structure is broken. Therefore, the microstructure of the plastically deformed layer of wheel material is a lamellar structure of ferrite lines and broken pearlite. Fatigue cracks are easy to initiate on the surface of wheel rollers and then propagate toward the interior material at a certain angle along the direction of the ferrite line (Figure 11a,b). Under the cyclic rolling contact load, the material above the crack fractures, resulting in the fractured material of the crack from the wheel roller surface to form wear debris (Figure 11). Then, pits are formed on the wheel roller surface, and the crack will continue to propagate toward the inner material along the direction of plastic deformation after fracture. Figure 11c is a schematic view of fatigue crack initiation and propagation of wheel material.

The plastic deformation lines of wheel specimens tend to develop toward the inner material under less tangential force. However, they tend to parallel the surface development of wheel specimens in the case of large tangential force. That is, as the tangential force increases, the plastic flow lines of the wheel specimen tend to become gradually parallel to the surface [13]. Therefore, the cracks on the wear surface with relatively large tangential force progress into the inner material at a certain depth, become parallel with the wheel roller surface, and then turn toward the surface or join with another crack; which allows the piece of material above the crack to become detached, resulting in surface spalling damage morphology (Figure 12a,b). This is similar to the observation of two typical surface fracture types (pitting and wear by spalling) in rolling fatigue tests by Sosnovskiy [22]. The spalling manifests in the separation of fine flakes or plates of the embrittled metal. A schematic view of the fatigue crack initiation and development of surface spalling damage is shown in Figure 12c.

The surface spalling damage morphology of the wheel (Figure 12b) is very similar to the surface morphology formed by the fracture of the crack material (Figure 11b), but the formation process is different. The similarity is that all the obvious pits are formed on the wheel roller surface. The difference is that the pits in Figure 11 are formed by the cyclic rolling contact, causing the crack to fracture and fall off during the propagation process; while the pits in Figure 12 are formed by the fatigue cracks turning and eventually propagating toward the wheel roller surface, which is easily distinguished in their formation schematic diagram (Figure 11c and Figure 12c).

The plastic deformation lines of the wheel specimen are nearly parallel to the surface under larger tangential force conditions, and the fatigue crack initiated on the wheel surface immediately propagates with a small angle parallel to the surface (Figure 13). The material above the crack is very thin, and turning to the surface causes the material to peel off and form worn debris, so that no significant pits are formed on the wheel roller surface. The above-mentioned process of crack propagation on the surface of wheel specimen causes the peeling damage. The schematic diagram for the development of peeling surface damage is shown in Figure 13c. It is known from the literature that the formation mechanism of spalling damage is very similar to that of peeling damage [30]. If the depth of a fatigue crack is relatively larger, once the material above the crack is removed, a significant pit is formed on the surface, which is called the surface spalling damage morphology (Figure 12c). If the depth of the fatigue crack is relatively small, the material above the crack is less likely to be broken and removed, and no significant pit is formed on the surface to form peeling morphology (Figure 13c).

In conclusion, the contact pressure and tangential force between the wheel and rail interface have a great influence on wear behavior and the tribo-fatigue damage mechanism of wheel material. According to the experimental data under different tangential forces and contact pressure conditions, the wear regime map and tribo-fatigue damage transition map have been proposed for the wheel material. It can be known through the tribo-fatigue damage mechanism map constructed in this study, that adjusting and controlling the load (contact pressure) and the interface friction force are effective measures for mitigating and preventing the tribo-fatigue damage of wheel material. It provides a certain theoretical basis for establishing design criteria for anti-tribo-fatigue damage of wheel materials. However, this study is purely experimental research. The friction, wear, and tribo-fatigue damage mechanism of wheel and rail materials will be theoretically analyzed in the future, based on damage mechanics and mechanical thermodynamic entropy.

## 5. Conclusions

The wear rate of the wheel specimen increases as the contact pressure and tangential force increases. According to wear rate of the wheel specimen under each testing condition, the wear mechanism map was established and divided into three regions of slight wear, severe wear, and destructive wear.The tribo-fatigue damage map of wheel material was established and divided into slight fatigue damage, fatigue damage, and severe fatigue damage regions, based on the analysis of the surface damage morphology and average crack length of the wheel roller under each experiment condition. In the slight fatigue damage region, the damage mechanism is slight fatigue damage and spalling; that of the fatigue damage region is dominated by surface fatigue cracks and peeling; and that of severe fatigue damage region is mainly severe surface fatigue cracks and peeling.The formation process of spalling damage was summarized based on the analysis of the test results, which was similar to the observations of two typical surface fracture types (pitting and wear by spalling) in rolling fatigue tests by Sosnovskiy [22]. Fatigue cracks are mainly initiated on the wheel surface. In the case of a small tangential force, some cracks propagate into material but fractured under cyclic rolling contact, and/or grow into inner material to a certain depth and then turn toward the surface to form material flaking. Under a large tangential force, some cracks always propagate parallel to the wheel roller surface.

## Figures and Tables

**Figure 1 materials-12-04138-f001:**
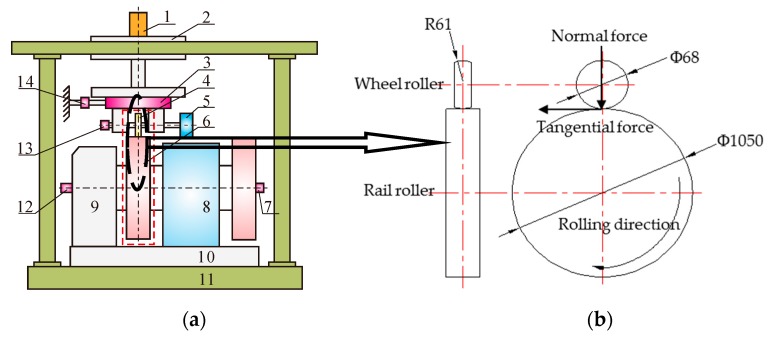
(**a**) the schematic and geometry of testing apparatus, **1**: Normal loading cylinder; **2**: Loading carriage; **3**: Vertical and longitudinal force collection unit; **4**: Wheel roller; **5**: Magnetic powder brake; **6**: Rail roller; **7**: Revolution speed transducer; **8**: ZQDR-204 DC motor; **9**: Gear box; **10**: Turning plate; **11**: Base plate; **12**: Optical shaft encoder; **13**: Revolution speed transducer; **14**: Lateral force transducer; (**b**) scheme size of wheel and rail rollers.

**Figure 2 materials-12-04138-f002:**
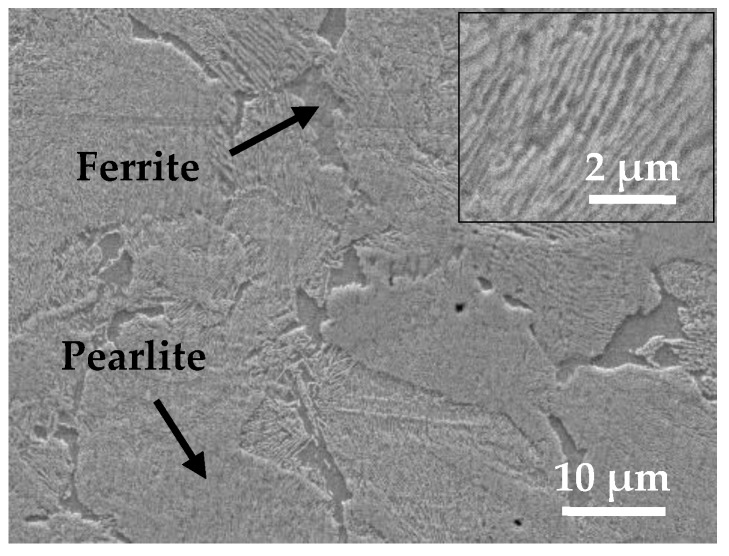
Microstructure of wheel material.

**Figure 3 materials-12-04138-f003:**
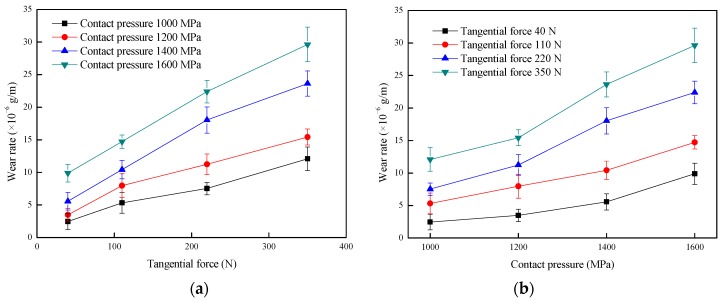
Effect of the tangential force and contact pressure on wear rate of wheel rollers; (**a**) wear rate vs. tangential force; (**b**) wear rate vs. contact pressure.

**Figure 4 materials-12-04138-f004:**
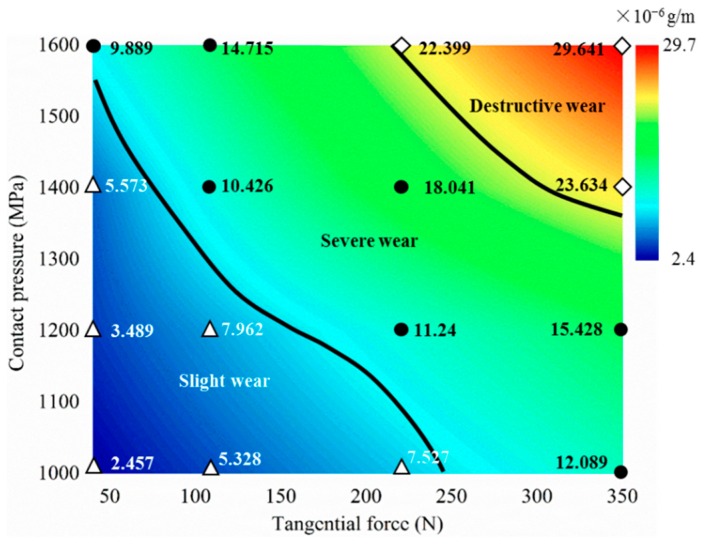
Wear regime map of wheel material.

**Figure 5 materials-12-04138-f005:**
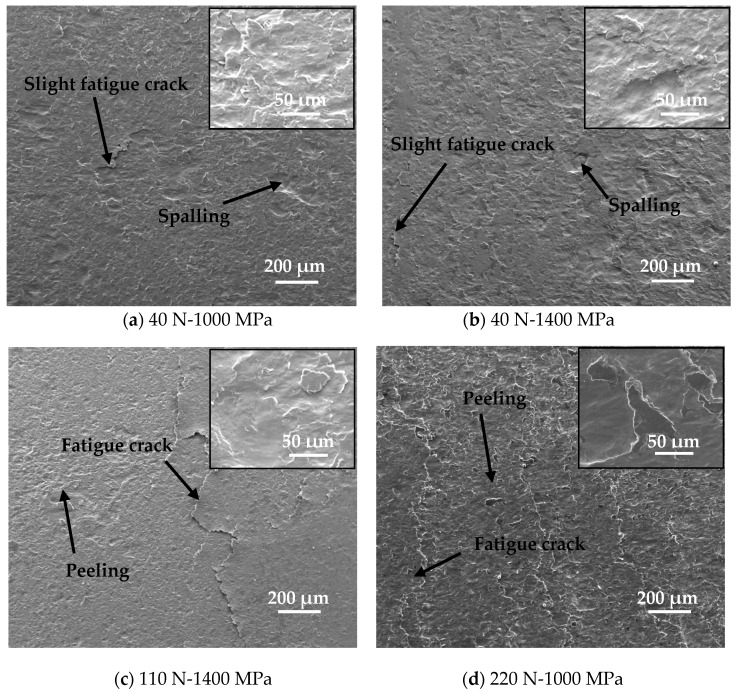
Surface damage morphologies of wheel material, (**a**) 40 N-1000 MPa; (**b**) 40 N-1400 MPa; (**c**) 110 N-1400 MPa; (**d**) 220 N-1000 MPa; (**e**) 220 N-1200 MPa; (**f**) 350 N-1200 MPa; (**g**) 220 N-1400 MPa; (**h**) 350 N-1600 MPa.

**Figure 6 materials-12-04138-f006:**
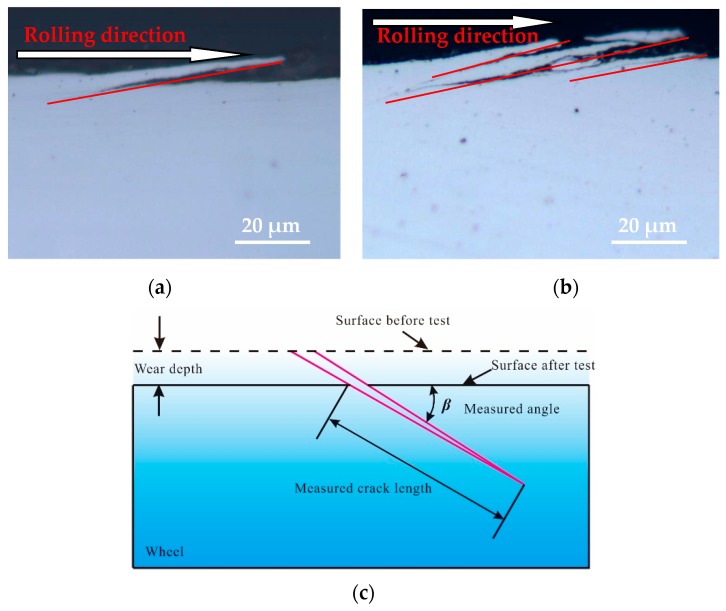
OM micrographs of sectional cracks of wheel roller, (**a**) single layer crack; (**b**) multi-layer cracks; (**c**) measurement of crack length and crack angle.

**Figure 7 materials-12-04138-f007:**
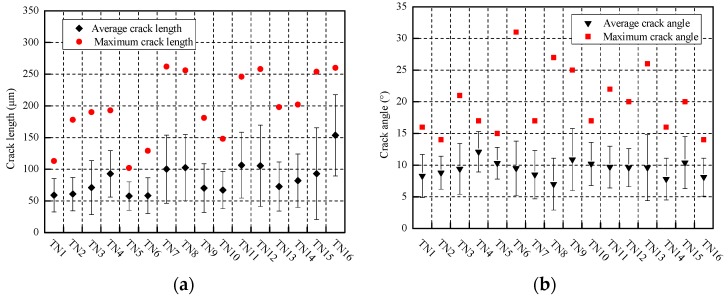
Crack length and crack angle of wheel rollers under different testing conditions, (**a**) crack length; (**b**) crack angle.

**Figure 8 materials-12-04138-f008:**
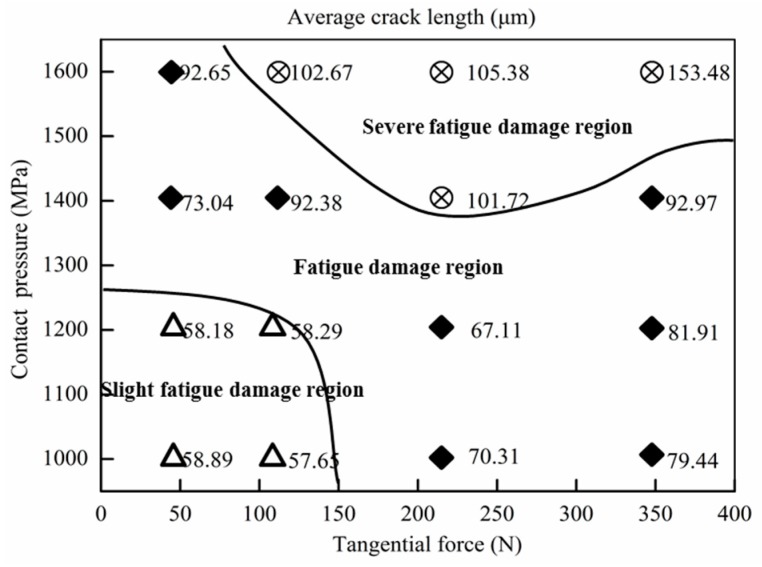
The tribo-fatigue damage transition map of wheel material.

**Figure 9 materials-12-04138-f009:**
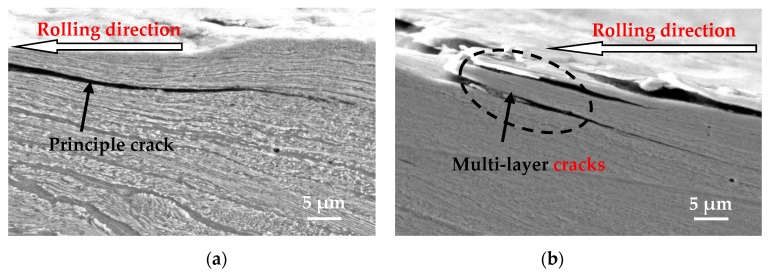
SEM micrograph of cracks of wheel rollers, (**a**) single principle crack; (**b**) multi-layer cracks; (**c**) crack broken; (**d**) sub-surface crack.

**Figure 10 materials-12-04138-f010:**
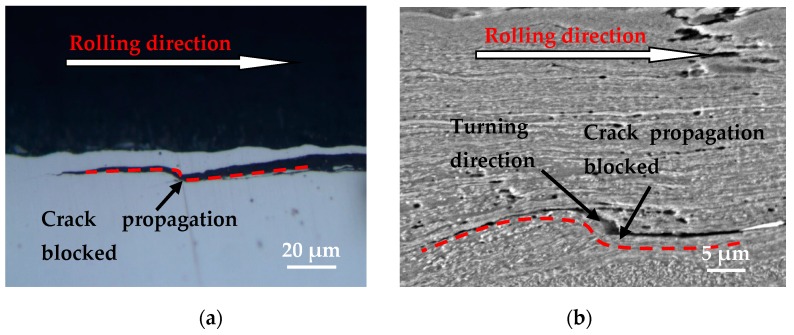
Crack propagation blocked and turning direction, (**a**) OM micrograph; (**b**) SEM micrograph.

**Figure 11 materials-12-04138-f011:**
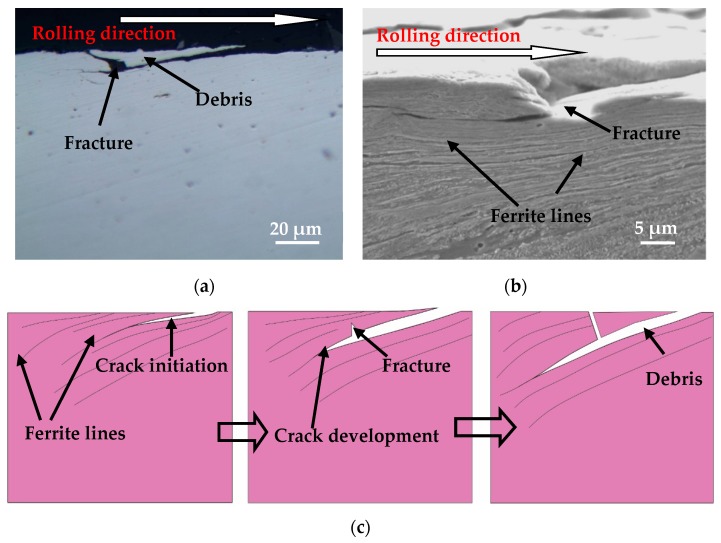
Fatigue crack of wheel material, (**a**) OM micrograph of fatigue crack; (**b**) SEM micrograph of fatigue crack; (**c**) schematic diagram of fatigue crack.

**Figure 12 materials-12-04138-f012:**
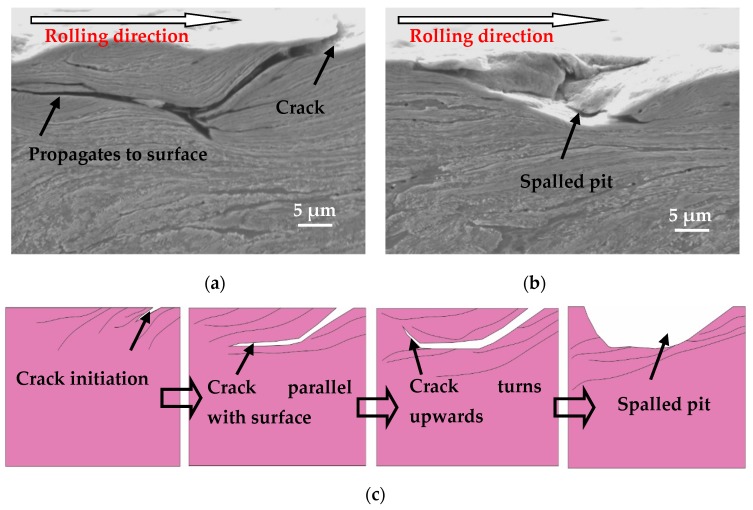
Spalling damage of wheel material, (**a**) OM micrograph of spalling; (**b**) SEM micrograph of spalling; (**c**) schematic diagram of spalling.

**Figure 13 materials-12-04138-f013:**
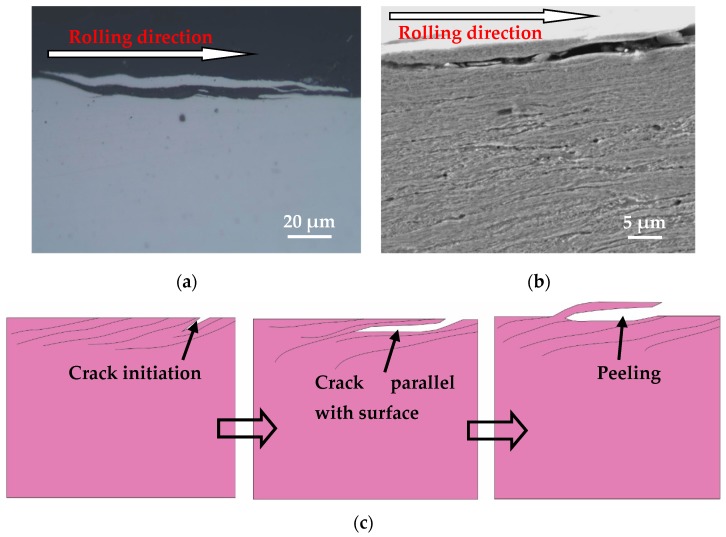
Peeling damage of wheel material, (**a**) OM micrograph of peeling; (**b**) SEM micrograph of peeling; (**c**) schematic diagram of peeling.

**Table 1 materials-12-04138-t001:** Chemical compositions and the main mechanical properties of wheel and rail rollers.

Materils	Chemical Composition (wt%)					The Main Mechanical Properties		
	C	Si	Mn	P	S	*σ*_b_ (MPa)	*δ*_10_(10%)	Hardness (HV_0.5_)
Wheel	0.56–0.60	≤0.40	≤0.80	≤0.020	≤0.015	≥880	≥10	280
Rail	0.65–0.75	0.10–0.50	0.80–1.30	≤0.025	0.008–0.025	≥900	≥10	300

**Table 2 materials-12-04138-t002:** Test matrix.

Test Number	Contact Pressure	Tangential Force	Test Number	Contact Pressure	Tangential Force
TN1	1000 MPa	40 N	TN9	1000 MPa	220 N
TN2	1200 MPa	40 N	TN10	1200 MPa	220 N
TN3	1400 MPa	40 N	TN11	1400 MPa	220 N
TN4	1600 MPa	40 N	TN12	1600 MPa	220 N
TN5	1000 MPa	110 N	TN13	1000 MPa	350 N
TN6	1200 MPa	110 N	TN14	1200 MPa	350 N
TN7	1400 MPa	110 N	TN15	1400 MPa	350 N
TN8	1600 MPa	110 N	TN16	1600 MPa	350 N

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
