# Peer review of "The Tribo-Fatigue Damage Transition and Mapping for Wheel Material under Rolling-Sliding Contact Condition"

_materials, 2019, doi:10.3390/ma12244138_

Round 1
Reviewer 1 Report
Please see attachment

Author Response
We appreciate greatly the valuable suggestions from the reviewer. We have carefully considered the comments and revised the manuscript accordingly.
Please see the attachment.

Reviewer 2 Report
Generally, three aspects are missing and not discussed:
1) rail geometry - presenting cross-sectional geometry (very detail) and possible unevenness, etc.
2) the system of loadings in contact area with with taking into account unevenness,
3) fatigue parameters of materials used for wheel versus durability.
Author Response

(The authors gave the same response as above.)

Reviewer 3 Report
The article ‘The tribo-fatigue damage transition and mapping for wheel material under rolling-sliding contact conditions’ by Chenggang He et al. is an article of acceptable quality and highlights important issues. In general, I recommend the article for publication in Materials journal with following comments.
The authors wrote that the chemical compositions and mechanical properties are provided in Table 1. But there are no mechanical properties, it should be supplemented. The methodology for determining the chemical composition and mechanical properties should be provided. Or indicate the references from which this data was obtained. Generally, the language needs revision. Errors appear in the text, e.g. ‘damge’ instead of ‘damage’. Old references should be replaced by more recent journal paper references. Conclusions are too general. It is necessary to compare the results obtained with the results from references.Author Response
We appreciate greatly the valuable suggestions from the reviewer. We have carefully considered the comments and revised the manuscript accordingly.
Please see the attachment.

Reviewer 4 Report
I consider the contribution of this investigation very interesting regarding the identification of wear mechanisms in the wheel/rail contact.
I recognized a few inaccuracies in the methodology and results in discussion and I don't quite agree with the authors' interpretation of the wear mechanism - that is why I suggest a major revision before publication in “Materials”.
My detailed comments:
- par. 1 – the aspect of fatigue of the wheel/rail contact is quite well recognized in the literature – this paragraph could contain a fragment about the main assumptions of spalling and shell initiation;
- par. 2 – and what about statistics? How many times was the test repeated for each pressure-tangential force configuration? Confidence intervals or at least standard deviations must be shown in the diagrams (fig. 3) – only in this way can we determine whether there are real differences between individual test parameters;
- Fig. 4 – why such wear values were used for the transition between individual regimes? Please explain or justify why 8 x 10^-6 g/m is already a severe regime and 20 x 10^-6 g/m is destruction?
- Fig. 5 – my interpretation of the fatigue mechanism based on available SEM images is different:
1) indeed, RCF traces are visible in the images and this is the main destructive mechanism, but:
2) on what basis did you find out that it is spalling? Did you measure the real temperature in the contact? Did you found the transformation of the microstructure into martensite?
3) what you call fatigue cracks is, in my opinion, a trace of very shallow "chipping" typical for shell not caused by thermal factors (pay attention please that RCF may have a thermal or non-thermal source);
- Fig. 7 – all forms of RCF can start this way – but please remember about the hydrodynamic factor, which has a very high impact on the evolution and dynamics of damage;
- Fig. 8 – pay attention please that according to the statistics on the average crack length, there is no difference between the vast majority of tests – It is also hard to assume some trend on the basis of fig. 8;
- p. 9, l. 238-244 – I agree that the cracking mechanism may be as you describe - but maybe sometimes the explanation is simpler - the directions of the cracks develop depending on the directions of the wheel slip relative to the rail? Tight curves, switches, etc.
- p. 9, l. 249-258 – pay attention please that you did not find martensite in the microstructure- according to many authors, this is a factor that determines the development of spalling;
Author Response

(The authors gave the same response as above.)

Round 2
Reviewer 1 Report
They did a good job incorporating my comments
Reviewer 4 Report
The authors' answers are satisfying - therefore I recommend this paper for publication in 'Materials'.